# Friction and Wear Characteristics of Microporous Interface Filled with Mixed Lubricants of M50 Steel at Different Loads

**DOI:** 10.3390/ma13132934

**Published:** 2020-06-30

**Authors:** Xiyao Liu, Zhiwei Lu, Hao Dong, Yan Cao, Xueming Qian

**Affiliations:** School of Mechatronic Engineering, Xi’an Technological University, Xi’an 710021, China; luzhiwei@xatu.edu.cn (Z.L); donghao@xatu.edu.cn (H.D.); caoyan@xatu.edu.cn (Y.C.); qianxueming@xatu.edu.cn (X.Q.)

**Keywords:** friction and wear, surface micropore, lubricant, M50 steel

## Abstract

Improving the tribological performance of M50 steel under the conditions of wide load range is of great significance. In this study, the interfacial structure of surface micropores filled with Sn–Ag–Cu or Sn–Ag–Cu/whisker carbon nanotubes (whiskerCNT) of M50 material was prepared by laser additive manufacturing and high-temperature infiltration. From 2 to 22 N, the lubrication characteristics of Sn–Ag–Cu and whiskerCNT in surface micropores of M50 was investigated. Results indicate that Sn–Ag–Cu can precipitate to the worn surface and form a lubricating layer, which has a good lubricating effect. Moreover, the flow behavior of Sn–Ag–Cu on a worn surface can play the role of crack healing. At higher load, the strength of the lubricating film is enhanced by whiskerCNT, which renders the lubricating film not vulnerable to premature rupture.

## 1. Introduction

M50 steel material has excellent thermal stability and anti-fatigue properties, and it is one of the main raw materials for manufacturing aviation spindle bearings [1,2,3]. Currently, the operation of aircraft bearings is developing more and more in the direction of heavy load and high speed; therefore, improving the tribological performance of M50 is particularly important, which is one of the research hotspots in the world [4,5,6].

Scholars achieved a series of achievements in this regard through research. Shi et al. [7] thoroughly investigated the impact of solid lubricant MoS_2_ on M50 friction properties at different temperatures, where the generation of the lubricating film containing the lubricant MoS_2_ was one of the main factors of the excellent lubricating properties of M50. However, if the temperature was too high, the structure of MoS_2_ as destroyed, and the lubricating effect was reduced. Researchers found that layered lubricant Ti_3_SiC_2_ can play a lubricating role in high-temperature conditions, as well as reduce the friction coefficient of M50 material [8], but this lubrication effect was not obvious in the middle- and low-temperature states. As can be concluded, a single solid lubricant struggles to prevent the M50 material from having outstanding friction and wear characteristics in a wide load range or temperature range. The enhancement effects of mixed lubricants TiO_2_ and graphene nanomaterials in terms of M50 friction reduction and wear resistance were studied experimentally at different temperatures [9]. The results state clearly that the friction and wear performance of M50 is distinguished in the wide temperature range. Composite solid lubricants SnS/ZnO were used to reduce the friction coefficient and wear rate of the material M50 in high-speed conditions [10]. The tribological characteristics of M50 were improved by the tribo-layer containing ZnO/SnS formed on the friction surface at high speed.

However, the preparation of the above friction materials is mainly based on powder metallurgy technology, where they are formed by mixing and sintering the lubricant with the metal matrix powder, which will reduce the strength of the friction material. Therefore, it is not conducive to self-lubricating M50 with good tribological performance in a wide load range. Adjusting the relationship between the lubricating property and the mechanical strength of friction materials is very important. Liu et al. [11] sintered a porous ceramic matrix friction material using a hole-making agent, but the porous size was not easy to control, and there was a certain degree of randomness. Our research team designed a surface microporous friction structure-filled lubricant, which could realize the unity of the lubricating performance and mechanical properties of the friction material [12].

Lubricants have an important influence on the tribological behavior of self-lubricating materials [13,14,15,16]. Sn–Ag–Cu alloy has good lubrication ability, and its bonding property with Fe is also good [17,18,19]; thus, Sn–Ag–Cu is suitable for improving the self-lubricating properties of M50. In addition, whisker carbon nanotubes (WhiskerCNT) with high strength can be used to improve material strength [20,21]. In view of the heavy-duty working conditions of M50 bearings, M50 material with surface micropores filled with Sn–Ag–Cu (MMS) or Sn–Ag–Cu/whiskerCNT (MMSC) was prepared by laser additive manufacturing (LAM) and high-temperature infiltration. The research focuses on the lubrication mechanism of whiskerCNT and Sn–Ag–Cu in surface micropores of M50 in a wide load range, which is very important for enhancing the tribological behavior of M50 material in the heavy-load field.

## 2. Materials and Methods

### 2.1. Lubricants and Matrix Materials

The elemental contents of M50 material are shown in Table 1. M50 spherical powders used to prepare M50, MMS, and MMSC were obtained by vacuum atomization. The principle of vacuum atomization method is that the alloy completes the melting, refining, and degassing process in the crucible of the vacuum chamber, and the refined metal liquid enters the atomizer through the diversion; then, the metal liquid flow is broken by high-pressure gas and dispersed to form a metal droplet in the atomization chamber, where, during the flight, the droplet is spheroidized and solidified to form the metal powder. The particle size range 20–50 μm was screened out. The micromorphology of the M50 spherical powder obtained by scanning electron micrograph (SEM) (JEOL Corporation, Tokyo, Japan) is shown in Figure 1a. The surface of the spherical powder was smooth, and the sphericity was good. The X-ray diffraction (XRD) (Rigaku Corporation, Tokyo, Japan) pattern in Figure 1a shows that M50 spherical powder is mainly composed of Fe_3_C and Fe. The results of previous studies indicated that the lubricant 50Sn40Ag10Cu was matched with the M50 matrix to provide excellent lubricating performance [22]. Vacuum atomization technology is also suitable for the preparation of 50Sn40Ag10Cu powder, and its particle size should be between 25 and 45 μm. The SEM morphology of 50Sn40Ag10Cu powder is shown in Figure 1b, where it can be seen that the surface quality of 50Sn40Ag10Cu powder is good. The XRD pattern (see Figure 1b) shows that Sn, Ag, Ag_3_Sn, and Cu_6_Sn_5_ were the main components, which also indicates that intermetallic reactions occurred during preparation. The microstructure of whiskerCNT is shown in Figure 1c, revealing whiskerCNT to be a short rod-shaped, hollow tubular structure. 

### 2.2. Preparation of Self-Lubricating Materials 

Three types of samples were prepared: pure M50 and self-lubricating materials MMS and MMSC. The preparation methods of the samples were LAM and infiltration technology. Previous studies [12] showed that the material has excellent mechanical and tribological properties with the following structure parameters of micropores: porosity 20%, depth 1 mm, and diameter 100 μm; the friction coefficient and wear rate could be reduced by about 20% and 40%, respectively. 

The preparation process of the sample was mainly divided into two steps. Firstly, the M50 preform with a surface microporous structure was prepared by LAM (LDM-8060 powder metal printer, Raycham Laser Technology Corporation, Nanjing, China), and the micropore area accounting for the entire friction surface was 3.5%. Table 2 lists the process parameters of LAM. Secondly, Sn–Ag–Cu/whiskerCNT mixed powders infiltrated the surface micropores of M50 preform using high-temperature melt infiltration, and the proportion of whiskerCNT powder in the mixed powders of Sn–Ag–Cu/whiskerCNT was 1 wt.%. The surface microporous M50 self-lubricating material was, thus, obtained. The parameters of the melt infiltration process are shown in Table 3. The preparation process of the surface microporous M50 self-lubricating material is shown in Figure 2.

### 2.3. Friction Tests

The device (Rtec Instruments, San Jose, CA, USA) used for the friction test is shown in Figure 3a. The contact mode for grinding was a ball disc contact, and the test motion method was operated under rotational mode (see Figure 3b). As a counterpart, the Si_3_N_4_ ball had a diameter of 6 mm, and its hardness (15 GPa) was much greater than that of M50 material, which meets the basic requirements for grinding pairs. The friction test load was variable, and the values were 2 N, 7 N, 12 N, 17 N, and 22 N. The test temperature and sliding velocity were 350 °C and 0.2 m/s, respectively. The air humidity in the laboratory was 35–55%. Under the same conditions, the friction test was performed three times to determine the dispersion of the measurement results. The dynamic friction coefficient was determined by measuring the friction force between the sample and the Si_3_N_4_ ball using the sensing system, and the ratio of the friction force to the vertical load applied on the sample surface was the friction coefficient. The friction test time was 80 min. A profilometer (Rtec Instruments, San Jose, CA, USA) was used to measure the profile of the wear scar and obtain the three-dimensional (3D) data map (Figure 3c). Then, along the path KK, the two-dimensional (2D) contour data (Figure 3d) of the cross-section of the wear scar was obtained, while the location selection of the cross-section of the wear scar was random. The product of the mean circumferential wear path and the area of the 2D profile curve was the wear volume.

## 3. Results and Discussion

### 3.1. The Interface of Micropores

Figure 4 shows the micro-morphology (see Figure 4a) and energy-dispersive spectroscopy (EDS) analysis patterns (see Figure 4b–f) near the micropores on the surface of MMSC. The light-gray substance in the middle is the filled lubricant Sn–Ag–Cu/whiskerCNT, and the dark-gray substance surrounding the Sn–Ag–Cu/whiskerCNT is the M50 matrix material, while there is no obvious gap at the boundary between the Sn–Ag–Cu/whiskerCNT and M50 matrix. Figure 5a shows the cross-section micromorphology at the micropore of theM50 self-lubricating material, where the dark-gray part on both sides is M50 material, and the light-gray part in the middle is the Sn–Ag–Cu/whiskerCNT filled in the micropores. In addition, at the boundary, M50 and Sn–Ag–Cu/whiskerCNT were integrated, which indicates that the combination form of the two materials was good. As can be seen from Figure 5b–f, the elemental distribution at the microporous boundary of the self-lubricating material was well distributed, where Sn, Ag, and Cu were the main components. Furthermore, the element diffusion phenomenon at the edges of micropores is presented, which is conducive to improving the bonding properties of M50 and Sn–Ag–Cu.

### 3.2. Friction Coefficient and Wear Rate

The mean friction coefficients and wear rates of M50, MMS, and MMSC were obtained via a friction test from 2 N to 22 N. As shown in Figure 6, material M50 had a greater friction coefficient and higher wear rate than the MMS or MMSC materials in the whole study load range, which implies that the excellent wear performance of M50 depended heavily on lubricants Sn–Ag–Cu and whiskerCNT. Figure 6a shows that, when the loads were 2–12 N, the friction coefficients of MMS were small, and the lowest friction coefficient of 0.12 was obtained at 12 N. When the applied load was 17 N or 22 N, the friction coefficient of MMS was larger, reaching 0.26 or 0.32, respectively. The trend of friction coefficient for MMSC with changing load was obviously different from that for MMS. In the whole load range, the friction coefficient of MMSC was lower than 0.2, which shows good friction reduction performance. Figure 6b presents the change regulation of the wear rates of M50, MMS, and MMSC at different loads. Obviously, self-lubricating MMS and MMSC materials both obtained the minimum wear rate when the load was 12 N, and, when the loads were 2 N and 7 N, the wear rates of MMS and MMSC were similar, fluctuating within the range of 15 × 10^−7^–17 × 10^−7^ mm^3^·N^−1^·m^−1^. When the load was increased to 17 N or 22 N, the wear rate of MMSC was reduced by about 30% compared to that of MMS. Based on the friction and wear behavior of the above materials, MMS and MMSC could improve the tribological characteristics of M50, and MMSC showed stable and distinguished anti-friction and wear resistance behavior throughout the studied load range, while MMS exhibited excellent tribological properties at lower loads. If the load was higher than 12 N, the friction coefficient and wear rate of MMS increased greatly.

### 3.3. Analysis of The Lubricating Mechanisms 

#### 3.3.1. The Worn Surfaces of M50

Figure 7 shows the micro-morphology of M50 on the wear scar obtained using an electron probe micro analyzer (EPMA) (JEOL Corporation, Tokyo, Japan). At 2 N, the worn surface of pure M50 presented a certain amount of grooves and abrasive debris (see Figure 7a), where plowing was the main wear mechanism. Some deeper grooves were found on the worn surface at 7 N, which may have been formed by the rolling of the Si_3_N_4_ ball on the M50 surface (see Figure 7b).

Figure 7c exhibits the EPMA morphology on the worn surface of M50 at 12 N, which was relatively smooth, whereas there was some tiny crushed debris, which showed that the worn surface was rolled flat at the load of 12 N. In addition, EDS analysis was implemented, and it can be seen from the EDS map in Figure 7e that the content of the O element was high, which also shows that there were some oxides on the worn surface. This is beneficial for improving the anti-friction and wear resistance of M50, which were some of the factors leading to M50 having a smaller friction coefficient and lower wear rate. At 17 N, the EPMA morphology of M50 was mainly dominated by deep grooves and plastic deformation (see Figure 7d), which could cause severe wear of the material. 

#### 3.3.2. The Worn Surfaces of MMS and MMSC

Figure 8 shows the micro-morphologies of the wear scars of MMS from 2 N to 17 N. At 2 N, the worn surface morphology (see Figure 8a) of MMS mainly appeared as some small grooves, and there were also some agglomerated particles, which may have been Sn–Ag–Cu precipitated from micropores due to heat and force. The friction surface shown in Figure 8b was obtained when the load was 7 N, showing large agglomerated particles on the surface of the wear scar, while there were also plows, and the wear mechanism was abrasive wear. At 12 N, the friction surface was relatively flat and there were island-like materials (Figure 8c). EDS analysis (see Figure 8e) showed that the main composition of these islands was Sn (20.1 wt.%), Ag (16.1 wt.%), Cu (2.19 wt.%), and O (9.80 wt.%), which clearly showed that Sn–Ag–Cu and oxides were the main components of isolated island materials. This confirms that the lubricant Sn–Ag–Cu was continuously precipitated to the worn surface of MMS, which was sufficient to form a continuous film of lubricant. Moreover, the excellent lubrication effect of the lubricating film enabled the material to obtain the lowest friction coefficient and wear rate (see Figure 6) at 12 N. At 17 N, deep grooves and plastic deformations were found on worn surface of MMS, and there were also some lubricant agglomerated particles. The main reason for this result may be that the lubricating film broke rapidly because of the increasing load, while the friction coefficient and wear rate of MMS (see Figure 6) increased sharply. 

Figure 9 shows the wear surface morphologies of MMSC under the conditions of 2 N, 7 N, 12 N, and 17 N. At 2 N, the worn surface shown in Figure 9a mainly presented small grooves and lubricant particles. This was due to the slight precipitation of Sn–Ag–Cu /whiskerCNT under the action of heat and force. At 7 N, Figure 9b presents the EPMA morphology of MMSC on the worn surface. Obviously, the morphology of MMSC was relatively flat, while there were also some small grooves and lubricant particles spread out on the worn surface of MMS. At 12 N, the worn surface of MMSC mainly presented a spreading lubricant layer, and the EDS pattern (see Figure 9e) of the worn surface indicated that elements Sn (19.6 wt.%), Ag (15.4 wt.%), Cu (2.05 wt.%), and O (8.2 wt.%) on the worn surface were detected, which may have been the diffused Sn–Ag–Cu from micropores. In addition, a large number of the C element (8.2 wt.%) on the worn surface indicated that whiskerCNT also precipitated from the micropores to worn surface. Figure 9d shows the EPMA morphology of MMSC when the load was 17 N. Compared with MMS, the friction surface morphology of MMSC was mainly flat, and there were no large plastic deformations, which characterized MMSC as still having excellent tribological properties at heavy load. The reason for this result can be inferred from two aspects. One is that the diffused Sn–Ag–Cu from micropores played a lubricating effect. The other is that whiskerCNT in the micropores migrated to the worn surface, which promoted the strength of the formed lubricating layer containing Sn–Ag–Cu to a certain extent, allowing the lubricating layer to withstand higher loads. 

#### 3.3.3. The Lubrication Mechanisms of Sn–Ag–Cu and WhiskerCNT

The lubrication mechanism of Sn–Ag–Cu and whiskerCNT on the surface needed further exploration at different loads. Figure 10 shows the micro-morphology and elemental distribution maps of the worn surface of MMSC at 17 N, while the maps shown in Figure 10c–e indicate that elements Ag, Sn, and Cu were distributed in sheets, which further confirmed the formation of a lubricating layer containing Sn–Ag–Cu. In addition, the presence of a large amount of oxygen also proves that some oxides also existed at heavy load.

In order to further analyze the lubrication mechanism, the cross-sectional morphologies and elemental distributions of MMSC on the worn surface at 17 N were analyzed. Figure 11a shows the cross-sectional micro-morphology of M50, where only one thin oxide film existed on the surface, which was insufficient in terms of lubricating performance compared with the self-lubricating material MMS or MMSC, and this was also validated according to the change regulation of friction coefficient and wear rate (see Figure 6). As shown in Figure 11b, where the cross-sectional micro-morphology of MMS on the worn scar is presented, a lubricating layer composed of Sn–Ag–Cu and whiskerCNT was obviously formed. While there were also some spalling behaviors of the lubricant layer, these were due to the lubricant film no longer being enough to withstand such a large load, resulting in the rupture of the lubricant film. In contrast, as shown in Figure 11c, the self-lubricating material MMSC formed a stable lubricating layer on the surface, and the thickness of the lubricating layer was increased compared with that of MMS. The elemental maps of the cross-section are shown in Figure 11d–h, where it can be seen that Sn, Ag, and Cu were abundantly accumulated on the surface of MMSC, which further confirmed the precipitation and flow spreading behaviors of Sn–Ag–Cu of MMSC.

X-ray photoelectron spectroscopy (XPS) (TMO Corporation, Waltham, MA, USA) was performed on the worn surface of MMSC, and the XPS spectra are shown in Figure 12. The XPS spectrum in Figure 12a shows Fe 2*p* peaks at 710 eV and 724 eV, which indicates the generation of Fe_2_O_3_ on the friction surface. As shown in Figure 12b, the positions of the Sn 3*d* peaks were at 486 eV and 494 eV, which indicated the presence of SnO_2_ on the friction surface of MMSC. The XPS spectrum (see Figure 12c) shows that the Ag peaks were at 367.9 eV and 373.9 eV, which confirms that Ag was agglomerated on the friction surface. The XPS spectrum of Cu 2*p* (see Figure 12d) reveals that the peaks were located at 933 eV and 953 eV, indicating that Cu on the worn surface was oxidized to CuO. In addition, the O 1*s* also had peaks (529–531.34 eV), which indicates that a large amount of oxidation reactions occurred during the friction process. Related research [23,24,25] indicates that oxides of Fe or Sn–Ag–Cu have distinguished lubricating properties, with a positive effect on improving the friction and wear performance. 

Based on the above analysis, the lubricant Sn–Ag–Cu had an outstanding lubrication effect, while whiskerCNT also migrated to the friction surface along with the Sn–Ag–Cu during friction. In order to investigate the mechanism of whiskerCNT in the lubrication of MMSC, field-emission scanning electron microscopy (FE-SEM) was used to evaluate the worn surface, and the test results are shown in Figure 13a. Obviously, whiskerCNT was wrapped by the lubricant Sn–Ag–Cu, forming a mixture of Sn–Ag–Cu and whiskerCNT. At lower load, Sn–Ag–Cu was the dominant factor in lubrication. When the load was high, whiskerCNT wrapped in the lubricating layer could effectively improve the bearing capacity of the lubricating film because of its high strength, such that the self-lubricating material MMSC could still keep its distinguished tribological performance at higher load. Figure 13b shows the Raman spectrum of MMSC on the worn surface, identifying the presence of iron oxides and whiskerCNT, which further verified the existence of whiskerCNT on the friction surface of MMSC. 

Figure 14a,b present the stress cracks at the edge of the wear scar of M50 at 17 N, and the widths of these cracks were measured to be about 188.6 nm and 179.6 nm, while the widths of cracks on the friction surfaces of MMS (see Figure 14c,d) were 49.5 nm, 46.9 nm, 45.3 nm, and 35.7 nm. After calculation, the width of the friction surface crack of MMS was reduced by approximately 74% with respect to that of M50, which indicates that the self-lubricating MMS had a crack self-repairing behavior on the friction surface. Related research [26] found that, if a material has two specific characteristics, the material can be used as a healing agent to help the damaged material achieve self-repairing of cracks; one is excellent flowability, while the other is the ability to adhere to the damaged material and fill the crack volume. Sn–Ag–Cu exhibits a molten state at high temperature, which can precipitate from the surface micropores under the combined influence of heat and force. Sn–Ag–Cu can also flow and spread on the friction surface because of its excellent extensibility, satisfying the condition of becoming a healing agent. The flow spreading state and crack-filling behavior of Sn–Ag–Cu can be clearly observed from Figure 14c,d. In addition, the results of EDS analysis at the crack are shown in Table 4, where the elemental contents of Sn, Ag, Cu, and O were found to be higher, which further confirmed that Sn–Ag–Cu had the function of repairing the crack.

Although the self-lubricating material MMS had the ability of self-repairing cracks, as shown in Figure 15, there was crack aggregation behavior at the edge of the wear scar at higher loading. This is because, under higher load, the lubricating layer mainly composed of Sn–Ag–Cu had limited compression capacity, and the rupture behavior of the lubricating layer occurred under constant higher loads. The rupture of the lubricating layer would inevitably cause an improvement of friction coefficient and wear rate of MMS to a certain extent, which was also confirmed by Figure 6 and Figure 8. Figure 16a shows the crack aggregation state at the edge of the wear scar of MMSC, and the width of the crack is shown in Figure 16b. The crack aggregation at the edge of wear scar of MMSC was not obvious, as there were only a few cracks, and the crack width measurement of the worn surface of MMSC in Figure 16b was only 29.1 nm. Compared with MMS, the crack width of the worn surface of MMSC was greatly reduced by about 30%, mainly due to the action of the added whiskerCNT. WhiskerCNT is a short tubular structure with high tensile strength and elastic modulus, and it is wrapped in the lubricating layer, which inevitably strengthens the lubricating layer and reduces the degree of crack diffusion. Based on the above tests and result analyses, the combined lubrication behavior of Sn–Ag–Cu and whiskerCNT equipped the self-lubricating material MMSC with remarkable friction and wear characteristics in the wide load range.

The lubrication mechanism of Sn–Ag–Cu and whiskerCNT in surface micropores is presented in Figure 17. In the initial state, the micropore on the surface of MMSC is filled with Sn–Ag–Cu and whiskerCNT (see Figure 17a,b). As the friction progresses, Sn–Ag–Cu and whiskerCNT in the micropores precipitate to the worn surface, and a lubrication layer rich in Sn–Ag–Cu and whiskerCNT is formed due to the continuous spreading flow behavior of Sn–Ag–Cu and whiskerCNT on the worn surface. WhiskerCNT is wrapped by Sn–Ag–Cu (see Figure 17c,d). Furthermore, the flow behavior of Sn–Ag–Cu on the worn surface allows it to be used as a healing agent to repair the microcracks, which is beneficial to promote the friction and wear performance of M50 (see Figure 17e,f). More importantly, the addition of whiskerCNT strengthens the formed lubricating layer of MMSC, which prevents the lubricating layer from rupturing at high load, such that MMSC still maintains good wear-reducing and wear-resistant performance under loading conditions of more than 12 N.

## 4. Conclusions 

Materials M50, MMS, and MMSC were prepared using LAM and infiltration technology. The cooperative lubrication behavior of Sn–Ag–Cu and whiskerCNT in surface micropores was investigated using ball-on-disc tests from 2 N to 22 N under conditions characterized by a temperature of 350 °C and a sliding speed of 0.2 m/s. The main conclusions of the study can be expressed as follows: (1)MMSC had stable and distinguished tribological properties from 2 to 22 N owing to the lubricating effect of Sn–Ag–Cu and whiskerCNT.(2)Sn–Ag–Cu could precipitate to the worn surface via the combined influence of heat and force, and a lubricating layer with a good lubricating effect was formed on the worn surface.(3)The flow spreading behavior of Sn–Ag–Cu could repair the stress cracks on the worn surface to some extent, which further enhanced the tribological properties of the M50 material.(4)Under heavy loads higher than 12 N, the lubricating layer formed on the friction surface produced a large number of crack aggregation behaviors, which eventually caused the lubricating layer to be destroyed.(5)WhiskerCNT enhanced the strength of the lubricating layer, which could greatly reduce the damage of the lubricating layer under loads heavier than 12 N.

## Figures and Tables

**Figure 1 materials-13-02934-f001:**
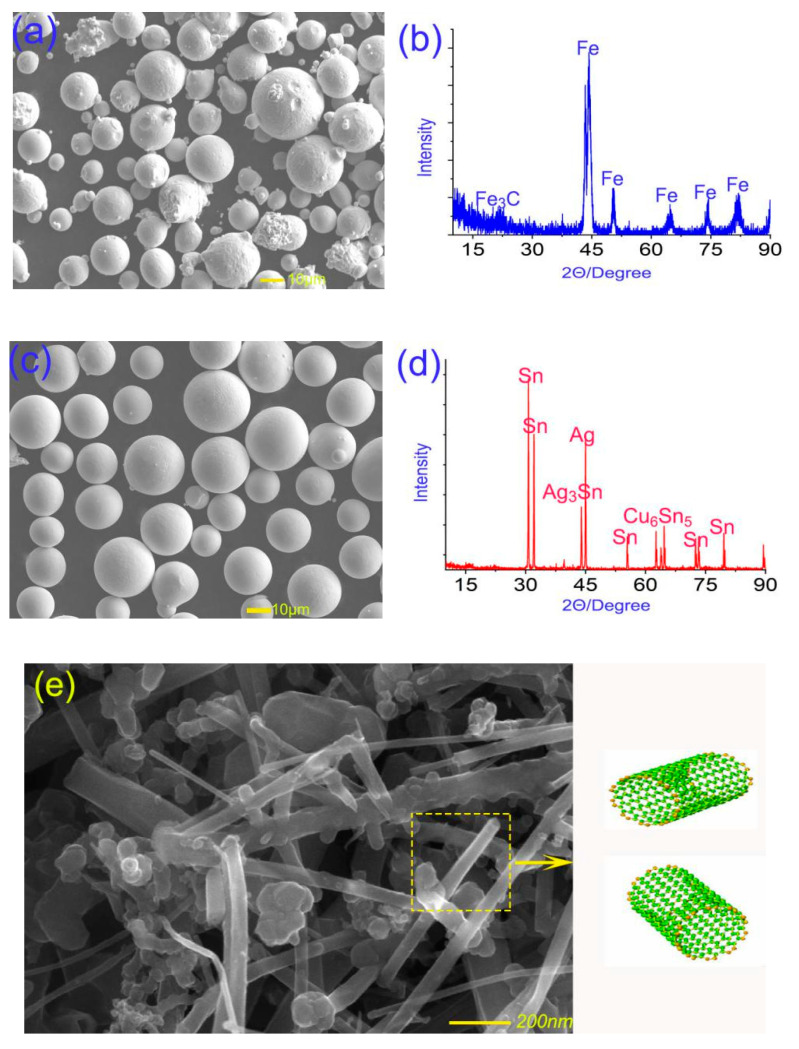
Micro-morphologies and X-ray diffraction (XRD) maps of M50 (**a**,**b**) and Sn–Ag–Cu spherical powder (**c**,**d**); the micro-morphology and structure of whisker carbon nanotubes (whiskerCNT) (**e**).

**Figure 2 materials-13-02934-f002:**
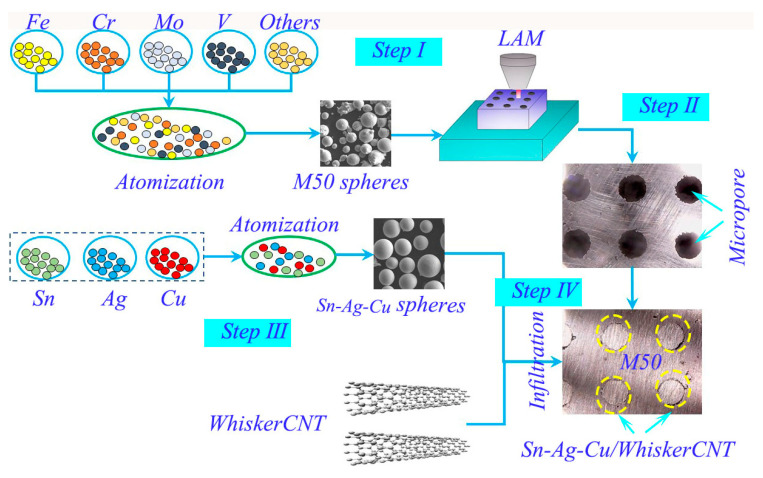
The flow chart for the preparation process of surface microporous M50 self-lubricating material.

**Figure 3 materials-13-02934-f003:**
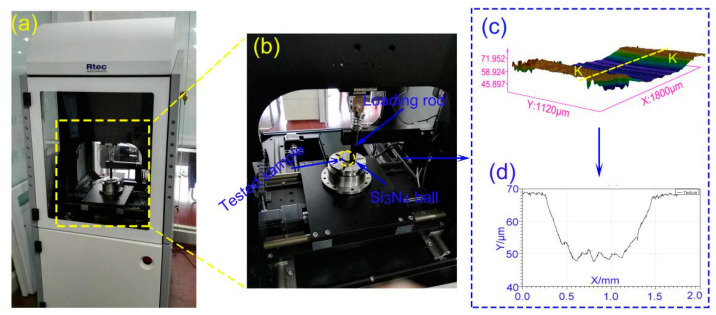
(**a**) The tribometer made by Rtec; (**b**) photos of friction test; (**c**) three-dimensional (3D) profile of wear scar; (**d**) two-dimensional (2D) profile of cross-section along the path KK.

**Figure 4 materials-13-02934-f004:**
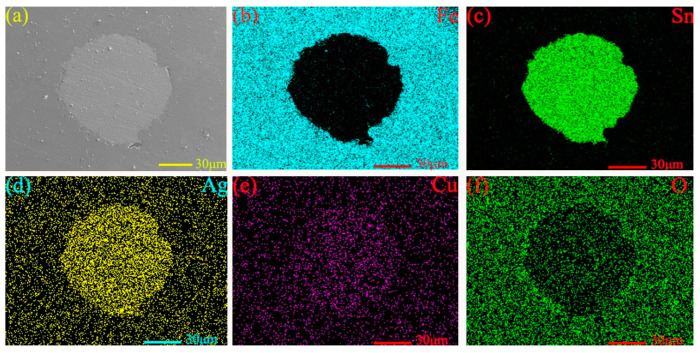
Micro-morphology (**a**) and energy-dispersive spectroscopy (EDS) maps (**b**–**f**) of the micropore on the surface.

**Figure 5 materials-13-02934-f005:**
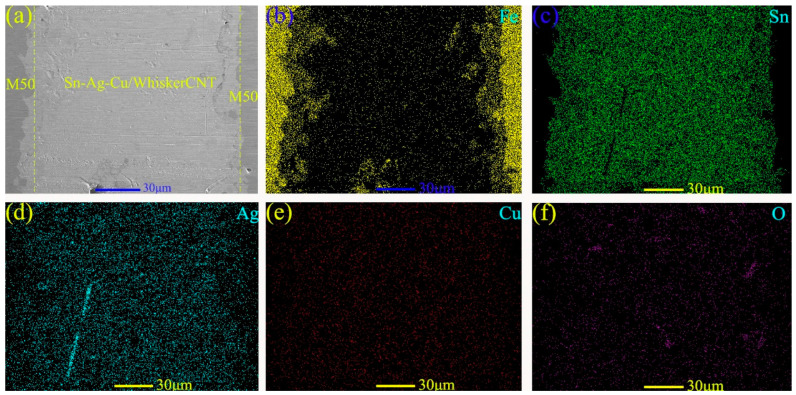
The cross-section micro-morphology (**a**) and EDS maps (**b**–**f**) of the micropore.

**Figure 6 materials-13-02934-f006:**
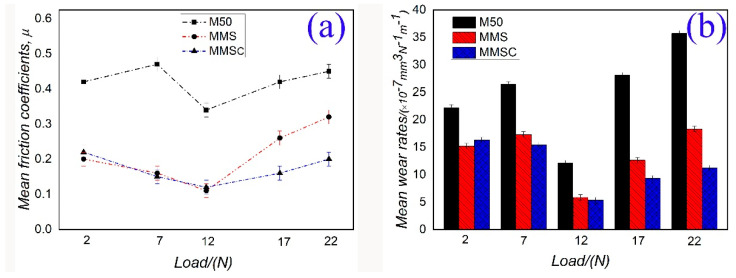
The mean friction coefficients (**a**) and wear rates (**b**) of M50, surface micropores filled with Sn–Ag–Cu (MMS), and surface micropores filled with Sn–Ag–Cu/whiskerCNT (MMSC).

**Figure 7 materials-13-02934-f007:**
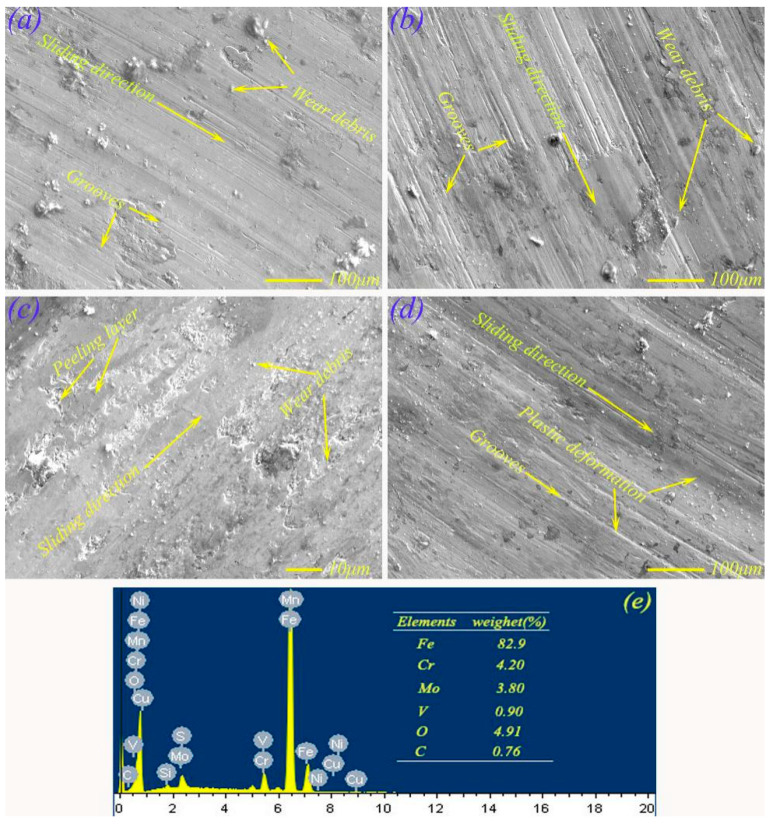
Electron probe micro analyzer (EPMA) morphologies of wear scars of M50 under different load conditions: (**a**) 2 N, (**b**) 7 N, (**c**) 12 N, (**d**) 17 N; EDS map at 12 N (**e**).

**Figure 8 materials-13-02934-f008:**
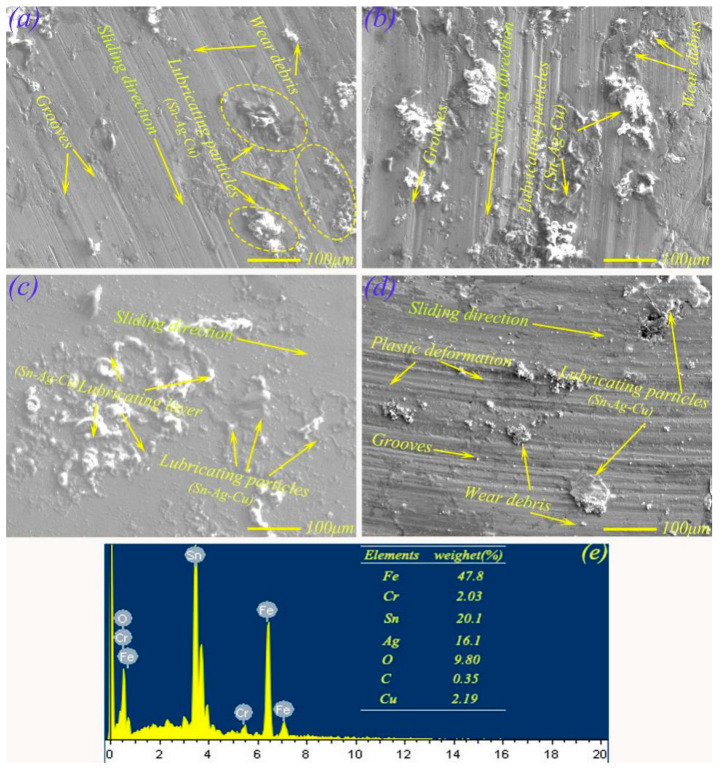
EPMA morphologies of the wear scars of MMS under different load conditions: (**a**) 2 N, (**b**) 7 N, (**c**) 12 N, (**d**) 17 N; EDS map at 12 N (**e**).

**Figure 9 materials-13-02934-f009:**
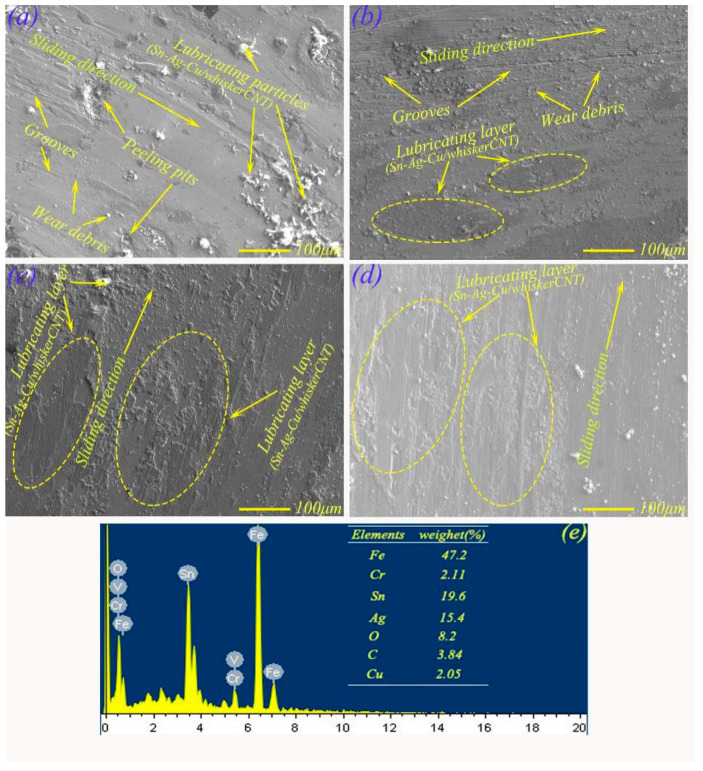
EPMA morphologies of the wear scars of MMSC under different load conditions: (**a**) 2 N, (**b**) 7 N, (**c**) 12 N, (**d**) 17 N; EDS map at 12 N (**e**).

**Figure 10 materials-13-02934-f010:**
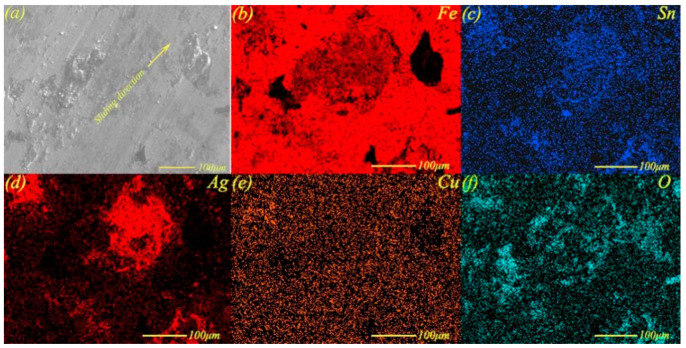
Micro-morphology (**a**) and elemental distribution maps (**b**–**f**) of worn surface of MMSC at 17 N.

**Figure 11 materials-13-02934-f011:**
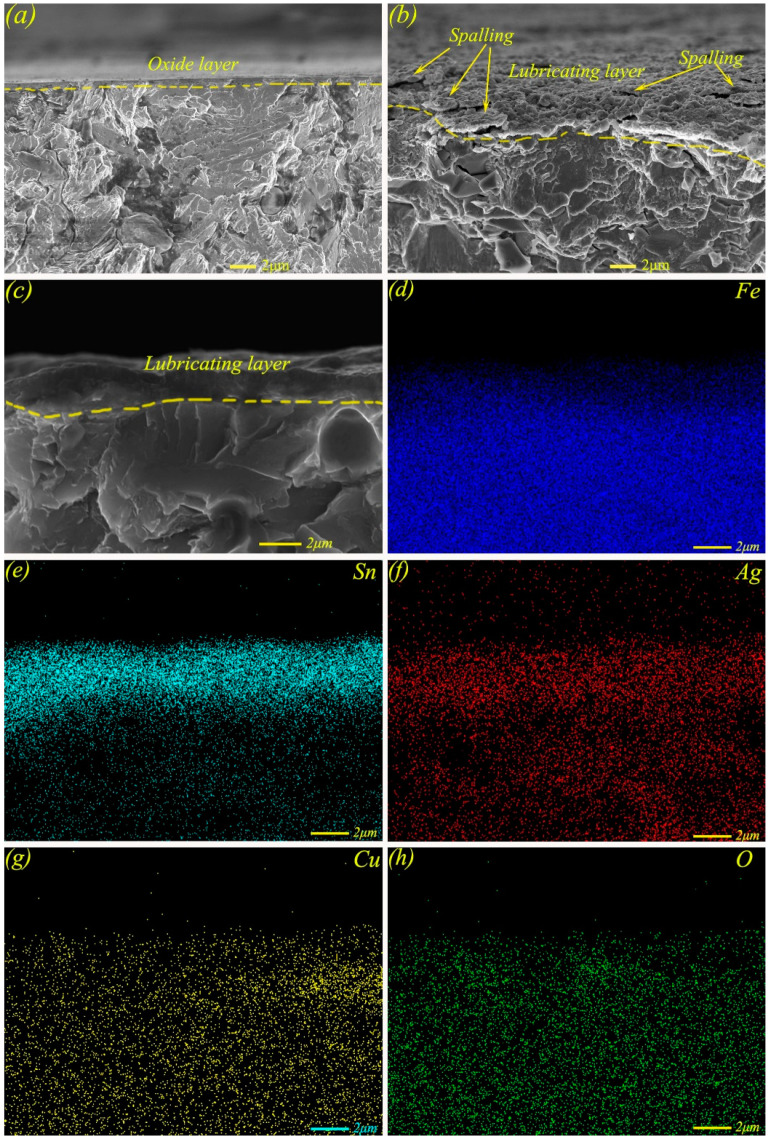
Micro-morphologies at the cross-section of wear scars of M50 (**a**) and MMS (**b**) at 17 N; micro-morphology (**c**) and elemental distribution maps (**d**–**h**) at the cross-section of the wear scar of MMSC at 17 N.

**Figure 12 materials-13-02934-f012:**
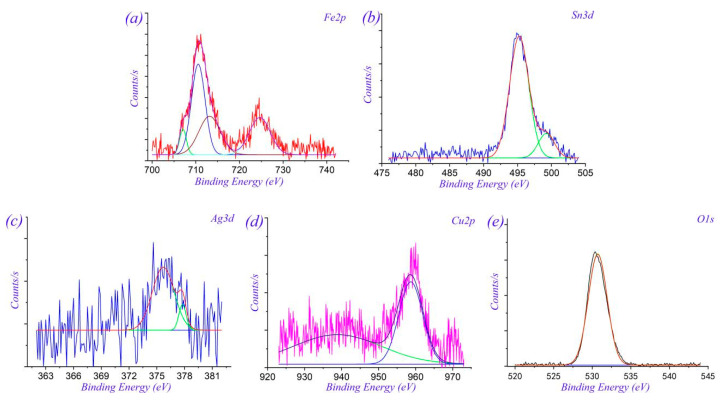
X-ray photoelectron spectroscopy (XPS) spectra of elements (**a**) Fe 2*p*, (**b**) Sn 3*d*, (**c**) Ag 3*d*, (**d**) Cu 2*p*, and (**e**) O 1*s* on the worn surface of MMSC.

**Figure 13 materials-13-02934-f013:**
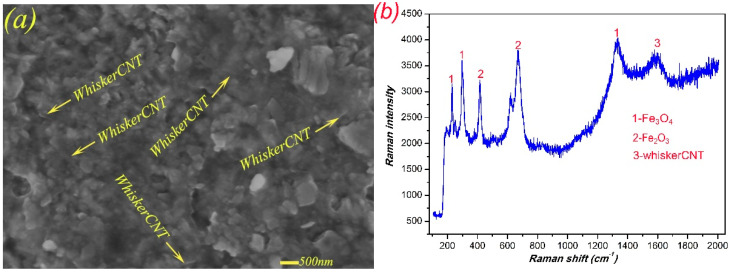
The micro-morphology (**a**) and Raman spectrum (**b**) of the worn surface of MMSC.

**Figure 14 materials-13-02934-f014:**
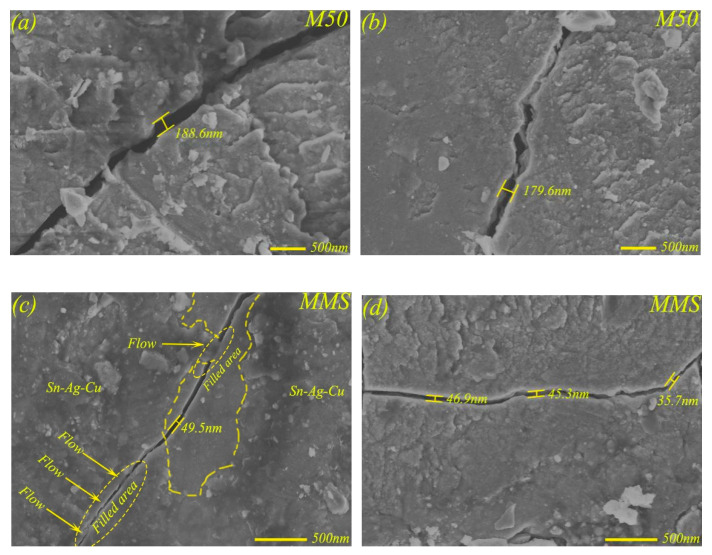
The stress cracks on worn surfaces of M50 (**a**,**b**) and MMS (**c**,**d**).

**Figure 15 materials-13-02934-f015:**
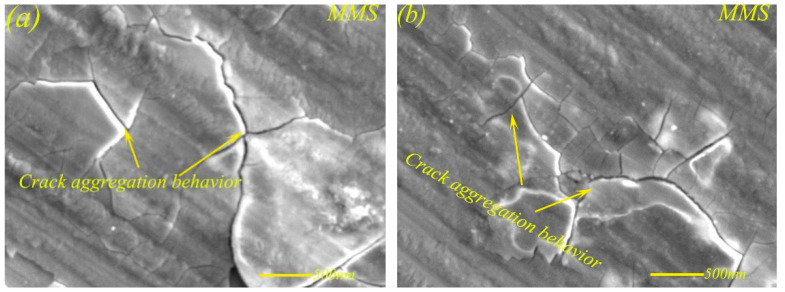
Crack aggregation behavior (**a, b**) at the edge of the wear scar of MMS at 17 N.

**Figure 16 materials-13-02934-f016:**
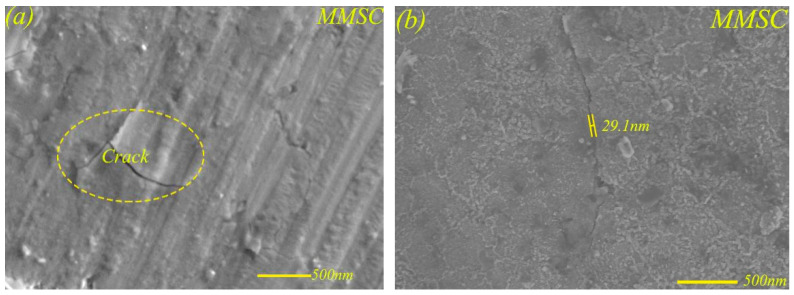
Crack aggregation (**a**) at the edge of the wear scar and the crack width (**b**) of MMSC at 17 N (**b**).

**Figure 17 materials-13-02934-f017:**
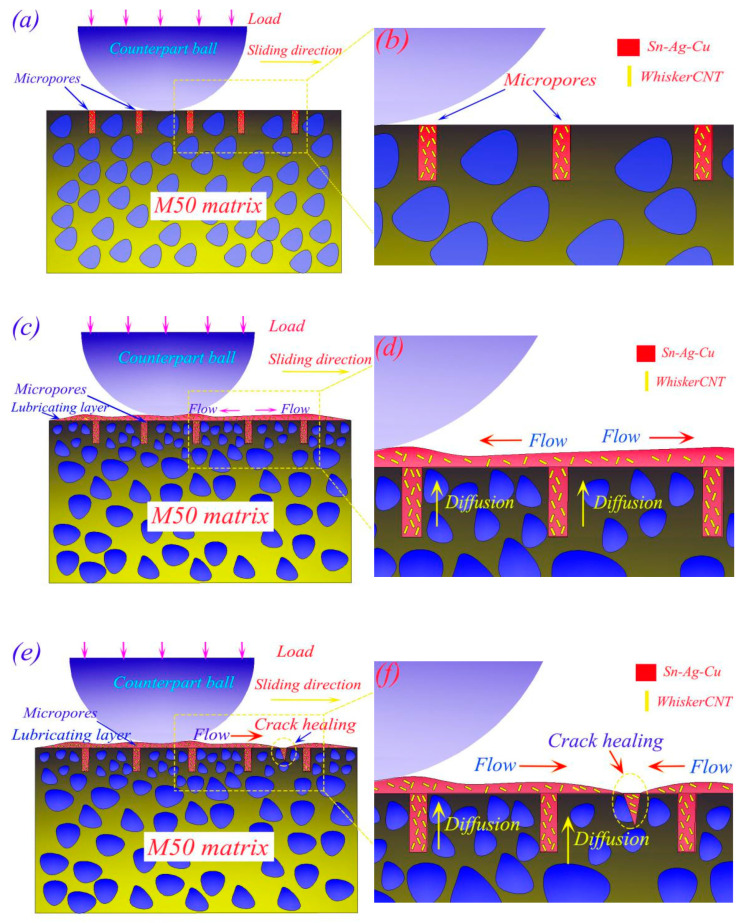
Schematic diagram of the lubrication mechanism of Sn–Ag–Cu and whiskerCNT in micropores: (**a**,**b**) the sample MMSC before friction, (**c**,**d**) the formation of lubricating layer, (**e**,**f**) the crack healing behavior.

**Table 1 materials-13-02934-t001:** The main elements of M50 steel (wt.%).

Elements	Cr	Si	V	Ni	C	Cu	Mo	Mn	Fe
Content	4.00	0.30	1.00	0.20	0.75	0.15	4.2	0.35	Balance

**Table 2 materials-13-02934-t002:** Process parameters of laser additive manufacturing (LAM).

**LAM**	**Working Pressure**	**Powder Transport Capacity**	**Protective Gas**	**Laser Power**
	3–5 mbar	5–10 g/min	Ar	2000 W

**Table 3 materials-13-02934-t003:** Process parameters of infiltration.

**Infiltration**	**Melting Temperature**	**Heating Power**	**Degree of Vacuum**	**Change of Pressure**
	500–650 °C	60–70 kW	0.85–0.93 Pa	0.70–0.80 MPa

**Table 4 materials-13-02934-t004:** Elemental contents at the crack of the worn surface (wt.%).

Elements	C	Cr	Fe	Sn	Ag	Cu	O
Contents	0.31	1.93	47.11	22.3	14.3	2.01	10.2

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
