# Peer review of "Friction and Wear Characteristics of Microporous Interface Filled with Mixed Lubricants of M50 Steel at Different Loads"

_materials, 2020, doi:10.3390/ma13132934_

Round 1
Reviewer 1 Report
What is the aim of the paper?
There is in line 67:
The element contents of M50 material are shown in Table 1, and the corresponding contents of elements Cr, Si, V, Ni, C, Cu, Mo, Mn and Fe are 4.00: 0.30: 1.00: 0.20: 0.75 : 0.15: 4.2: 0.35: 89.05.
Should be: The element contents of M50 material are shown in Table 1.
Line 69:
It would be useful to include a few sentences explaining the vacuum atomization method, because the article is also for laymen.
Figure 1 – the XRD analysis patterns should be placed in the separate Figure otherwise they obscure the readability of others.
Line 97:
It will be better to cite these values.
Table 2
This table must be either split into two tables or rearranged, with the appropriate column names selected. The current use of bold confuses, is it probably used incorrectly?
There is in Line 89:
Three types of samples need to be prepared: pure M50 and self-lubricating materials MMS and…
Should be:
Three types of samples were prepared: pure M50 and self-lubricating materials MMS and…
Line 113 - There is:
and the test motion method is rotary friction (See Fig.3a-b).
could be better:
operating under rotational mode (See Fig.3a-b)
Line 120: to reduces the test errors. (???)
Rather, to determine the dispersion of the measurement results?
The dynamic friction coefficient of friction experiment… - it needs to be explained how this coefficient was determined.
How long was/were wear test/tests – it needs to be given the number of rotates or time or friction path length.
Examples of friction coefficient as a function of time (even for a short time period) for each of the test loads could be useful to show the dynamics of friction coefficient variability.
Line 337 there is:
surface micropores can be described in Fig.17.
could be better:
surface micropores is presented in Fig.17.
Figure 17 – better resolution is needed.
Line 356 there is:
under the condition of 350oC-0.2m/s.
should be:
under conditions characterized by the temperature of 350oC and sliding speed of 0.2m/s.
It would be useful to compare the obtained friction coefficient and wear rate values with those obtained for a similar material combination and lubrication conditions or for another material combination, but similar conditions of loading, sliding speed, temperature, humidity and lubrication.
It would also be useful to indicate future research directions.
Author Response
Response to Reviewer 1 Comments
Point 1: What is the aim of the paper?

Response 1: Thank you for your advice. In view of the heavy-duty working conditions of M50 bearings, M50 self-lubricating material with surface micropores filled in Sn-Ag-Cu or Sn-Ag-Cu/whiskerCNT is prepared by laser additive manufacturing and high-temperature infiltration. The research focuses on the lubrication mechanism of whiskerCNT and Sn-Ag-Cu in surface microporous of M50 in a wide load range. This is very important to enhance the tribological behavior of M50 material in heavy load field. We also give a detailed explanation in this paper (In line 54-58).
Point 2: There is in line 67: The element contents of M50 material are shown in Table 1, and the corresponding contents of elements Cr, Si, V, Ni, C, Cu, Mo, Mn and Fe are 4.00: 0.30: 1.00: 0.20: 0.75 : 0.15: 4.2: 0.35: 89.05. Should be: The element contents of M50 material are shown in Table 1.
Response 2: Thank you for your advice. We have revised it according to your suggestion (In line 62).
Point 3: Line 69: It would be useful to include a few sentences explaining the vacuum atomization method, because the article is also for laymen.
Response 3: Thank you for your advice. In this paper, we have explained the basic principle of vacuum atomization method (Line 63-68).
Point 4: Figure 1 - the XRD analysis patterns should be placed in the separate Figure otherwise they obscure the readability of others.
Response 4: Thank you for your advice. We have modified this according to your suggestions.
Point 5: Line 97: It will be better to cite these values.
Response 5: Thank you for your advice. We have elaborated on the quoted content (Line 89-93).
Point 6: Table 2. This table must be either split into two tables or rearranged, with the appropriate column names selected. The current use of bold confuses, is it probably used incorrectly?
Response 6: Thank you for your advice. We have split Table 2 into two tables.
Point 7: There is in Line 89: Three types of samples need to be prepared: pure M50 and self-lubricating materials MMS and…
Should be: Three types of samples were prepared: pure M50 and self-lubricating materials MMS and…
Response 7: Thank you for your advice, and we fully agree with you. We have modified this statement according to your suggestion (Line 88).
Point 8: Line 113 - There is: and the test motion method is rotary friction (See Fig.3a-b).
could be better: operating under rotational mode (See Fig.3a-b)
Response 8: Thank you for your advice, and we fully agree with you. We have revised the statement according to your suggestion (Line 111).
Point 9: Line 120: to reduces the test errors. (???) Rather, to determine the dispersion of the measurement results?
Response 9: Thank you for your advice. We apologize for this inappropriate statement, and we have carefully revised the statement based on your suggestions (Line 116-117).
Point 10: The dynamic friction coefficient of friction experiment… - it needs to be explained how this coefficient was determined. How long was/were wear test/tests – it needs to be given the number of rotates or time or friction path length. Examples of friction coefficient as a function of time (even for a short time period) for each of the test loads could be useful to show the dynamics of friction coefficient variability.
Response 10: Thank you for your advice, and we fully agree with you. The friction coefficient is determined by measuring the friction force between the sample and Si3N4 ball, and the ratio of the friction force to the vertical load applied on the sample surface is the friction coefficient. The friction test time is 80min. We also give detailed explanations in the paper (Line 116-119).
Point 11: Line 337 there is: surface micropores can be described in Fig.17.
could be better: surface micropores is presented in Fig.17.
Response 11: Thank you for your advice, and we fully agree with you. We have modified this statement according to your suggestion (Line 323-324).
Point 12: Figure 17 - better resolution is needed.
Response 12: Thank you for your advice. We have revised the Figure 17 to improve its clarity.
Point 13: Line 356 there is: under the condition of 350oC-0.2m/s.
should be: under conditions characterized by the temperature of 350oC and sliding speed of 0.2m/s.
It would be useful to compare the obtained friction coefficient and wear rate values with those obtained for a similar material combination and lubrication conditions or for another material combination, but similar conditions of loading, sliding speed, temperature, humidity and lubrication.
It would also be useful to indicate future research directions.
Response 13: Thank you for your advice, and we fully agree with you. We have modified this statement according to your suggestion (Line 342-344).

Reviewer 2 Report
In aerospace applications the rolling bearings must face extreme temperatures. In such conditions, liquid lubricants can not withstand and must be replaced by solid lubricants. In the last years many researchers aborded this subject, but new solutions could offer better results.
Laser additive manufacturing and high-temperature infiltration were used to fill in the micropores of M50 samples with Sn-Ag-Cu (MMS) and Sn-Ag-Cu/whiskerCNT (MMSC). These samples ran on ball-on-disk tribometer against balls of silicon nitride at constant speed and temperature (0.2m/s and 3500C, respectively) and various applied load (2N, 7N, 12N, 17N, and 22N). Friction and wear of M50 samples impregnated with solid lubricants were analyzed by different techniques: scanning electron micrograph (SEM), energy dispersive spectroscopy (EDS), profilometry, XPS, and Raman spectrograph. Wear modes were studied by analysis of SEM images and a lubrication mechanism was proposed for both Sn-Ag-Cu (MMS) and Sn-Ag-Cu/whiskerCNT (MMSC) solid lubricants. Results proved that MMSC provide better resistance to wear at high loads (over 12 N), due to whiskerCNT higher resistance.
This paper is original and the research was well conducted. I appreciate the scrupulosity of the authors in interpreting the abundant experimental results. Anyway, in my opinion there still are minor amendments that ought to be done before publication. Consequently, I have just few remarks.
1. At section 2.1, the percent of Sn-Ag-Cu (MMS) or Sn-Ag-Cu/whiskerCNT (MMSC) in M50, expressed as wt.%, must be indicated in Table 1.
2. Some testing parameters on ball-on-disk tribometer were chosen as reference: constant speed of 0.2m/s and a temperature of 3500C. I saw in previously published works by the same authors that 3500C was found to provide the minimum friction coefficient. Also, 12 N seems to provide the least friction coefficient in this paper. The opinion of the authors about the optimum running parameters in a certain application is expected. The simulated speed and contact pressures are usually encountered in aerospace applications, or the research did not take into account this aspect?
3. Line 126, the wear volume is not computed using the radius, but considering the mean circumferential wear path.
4. Line 182, what is “the proper load”?! This expression ought to be explained, or replaced with a suited one.
5. Figures 17d and 17f: “fiow” is “flow”. Minor expression errors were also found, e.g., line 300 – “flow ability” is “flowability”, line 356 – “was investigated from 2N” is “was investigated by ball-on-disk tests from 2N”, line 369 – “under heavy load than 12N” is “under load heavier than 12 N”.
Author Response
Response to Reviewer 2 Comments
In aerospace applications the rolling bearings must face extreme temperatures. In such conditions, liquid lubricants can not withstand and must be replaced by solid lubricants. In the last years many researchers aborded this subject, but new solutions could offer better results.
Laser additive manufacturing and high-temperature infiltration were used to fill in the micropores of M50 samples with Sn-Ag-Cu (MMS) and Sn-Ag-Cu/whiskerCNT (MMSC). These samples ran on ball-on-disk tribometer against balls of silicon nitride at constant speed and temperature (0.2m/s and 350oC, respectively) and various applied load (2N, 7N, 12N, 17N, and 22N). Friction and wear of M50 samples impregnated with solid lubricants were analyzed by different techniques: scanning electron micrograph (SEM), energy dispersive spectroscopy (EDS), profilometry, XPS, and Raman spectrograph. Wear modes were studied by analysis of SEM images and a lubrication mechanism was proposed for both Sn-Ag-Cu (MMS) and Sn-Ag-Cu/whiskerCNT (MMSC) solid lubricants. Results proved that MMSC provide better resistance to wear at high loads (over 12N), due to whiskerCNT higher resistance.
This paper is original and the research was well conducted. I appreciate the scrupulosity of the authors in interpreting the abundant experimental results. Anyway, in my opinion there still are minor amendments that ought to be done before publication. Consequently, I have just few remarks.
Point 1: At section 2.1, the percent of Sn-Ag-Cu (MMS) or Sn-Ag-Cu/whiskerCNT (MMSC) in M50, expressed as wt.%, must be indicated in Table 1.
Response 1: Thank you for your advice. Firstly, M50 preform with surface microporous structure is prepared by LAM. Secondly, Sn-Ag-Cu/whiskerCNT mixed powders are infiltrated into the surface micropores of M50 preform by infiltration technology, and M50 materials with surface micropore filled in Sn-Ag-Cu or Sn-Ag-Cu/whiskerCNT are obtained. According to the friction interface structure and preparation process, we think that the micropore structure parameters can more accurately reflect the proportion of Sn-Ag-Cu/whiskerCNT. For a more accurate description, we also give the micropore area accounted for the entire friction surface, and the percentage is 3.5%. In addition, we also give the proportion (1wt.%) of whiskerCNT powder in the mixed powders of Sn-Ag-Cu/whiskerCNT. In the paper, we give detailed explanations (Line 91, 92, 96 and 99).
Point 2: Some testing parameters on ball-on-disk tribometer were chosen as reference: constant speed of 0.2m/s and a temperature of 350oC. I saw in previously published works by the same authors that 350oC was found to provide the minimum friction coefficient. Also, 12N seems to provide the least friction coefficient in this paper. The opinion of the authors about the optimum running parameters in a certain application is expected. The simulated speed and contact pressures are usually encountered in aerospace applications, or the research did not take into account this aspect?
Response 2: Thank you for your advice, and we fully agree with you. At present, the friction test has not yet reached the actual friction conditions of aviation bearings. We will focus on this research in the future.
Point 3: Line 126, the wear volume is not computed using the radius, but considering the mean circumferential wear path.
Response 3: Thank you for your advice. We apologize for this inappropriate statement, and we have carefully revised the statement based on your suggestions (Line 123).
Point 4: Line 182, what is “the proper load”?! This expression ought to be explained, or replaced with a suited one.
Response 4: Thank you for your advice. We apologize for this inappropriate statement, and we have revised the statement based on your suggestions (Line 176-177).
Point 5: Figures 17d and 17f: “fiow” is “flow”. Minor expression errors were also found, e.g., line 300 - “flow ability” is “flowability”, line 356 - “was investigated from 2N” is “was investigated by ball-on-disk tests from 2N”, line 369 - “under heavy load than 12N” is “under load heavier than 12 N”.
Response 5: Thank you for your advice. We are sorry for the inappropriate statements, and we have revised the statements based on your suggestions (Line 290, 335-337, 343 and 356).
